# Efficient Exploration in Multi-Agent Reinforcement Learning via Farsighted Self-Direction

**Tiancheng Lao**                                       *ltc22@mails.tsinghua.edu.cn*
*Department of Automation*
*Tsinghua University, Beijing, China*

**Xudong Guo**                                          *gxd20@mails.tsinghua.edu.cn*
*Department of Automation*
*Tsinghua University, Beijing, China*

**Mengge Liu**                                          *lmg22@mails.tsinghua.edu.cn*
*Department of Automation*
*Tsinghua University, Beijing, China*

**Junjie Yu**                                           *yjj23@mails.tsinghua.edu.cn*
*Department of Automation*
*Tsinghua University, Beijing, China*

**Yi Liu**                                              *yiliu@tsinghua.edu.cn*
*Department of Automation*
*Tsinghua University, Beijing, China*

**Wenhui Fan**                                          *fanwenhui@tsinghua.edu.cn*
*Department of Automation*
*Tsinghua University, Beijing, China*

**Reviewed on OpenReview:** *https://openreview.net/forum?id=NUV8THrLZC*

## Abstract

Multi-agent reinforcement learning faces greater challenges with efficient exploration compared to single-agent counterparts, primarily due to the exponential growth in state and action spaces. Methods based on intrinsic rewards have been proven to enhance exploration efficiency in multi-agent scenarios effectively. However, these methods are plagued by instability during training and biases in exploration direction. To address these challenges, we propose Farsighted Self-Direction (FSD), a novel model-free method that utilizes a long-term exploration bonus to achieve coordinated exploration. Since prediction error against individual Q-values indicates a potential bonus for committed exploration, it is taken into account in action selection to directly guide the coordinated exploration. Further, we also use clipped double Q-learning to reduce noise in the prediction error. We validate the method on didactic examples and demonstrate the outperformance of our method on challenging StarCraft II micromanagement tasks.

## 1 Introduction

Currently, multi-agent reinforcement learning (MARL) has been applied in multiple fields, such as autonomous driving (Wu et al., 2019) and traffic control (Zhang et al., 2019). In the field of games, where ample training data is more readily available, it has demonstrated its powerful potential, reaching human top-level performance in StarCraft II (Vinyals et al., 2019), Quake III Arena (Jaderberg et al., 2019), and Dota 2 (Berner et al., 2019). To facilitate the practical implementation of MARL, centralized training with

decentralized execution (CTDE) has been widely adopted as a MARL paradigm. It leverages global information during training, alleviating the non-stationarity problem (Gronauer & Diepold, 2022) while agents only rely on local observation history during execution, enabling distributed deployment. Under the CTDE framework, value decomposition methods have been continuously explored and refined(Sunehag et al., 2018; Rashid et al., 2018; Wang et al., 2021b; Xu et al., 2023), demonstrating excellent performance in numerous tasks.

Despite the current success, it has been found that with the simple $\epsilon$-greedy exploration strategy, these methods are sample-inefficient to solve complex tasks, especially those that require coordinated and persistent exploration (Wang et al., 2021b; Zheng et al., 2021). In the single-agent domain, there have been plenty of extensive studies, including uncertainty-oriented methods (Janz et al., 2019; Lee et al., 2021) and intrinsic motivation-oriented methods (Mohamed & Rezende, 2015; Pathak et al., 2017; Fox et al., 2018; Burda et al., 2019; Badia et al., 2020). Unfortunately, these methods cannot be directly applied to multi-agent systems due to the exponential growth of state and action spaces, partial observability, and the requirement for coordinated exploration (Zheng et al., 2021; Jiang et al., 2024).

To address the limitation, the intrinsic reward is designed as the prediction error of the individual Q-values (Zheng et al., 2021; Zhang & Yu, 2023) for value decomposition methods, promoting efficient coordinated exploration. To avoid excessive coupling between intrinsic reward generation and policy learning, these curiosity-driven methods typically employ a controller-explorer-predictor structure where the controller learns the policy used to interact with the environment and the explorer is only used during training to generate the intrinsic reward. The prediction error between the explorer and the predictor is used as the intrinsic reward for the controller to enhance exploration. Despite the impressive performance in the experiments, they still have two main drawbacks: (1) *instability during training.* The intrinsic reward varies as the training progresses; thereby, agents face non-stationary environmental dynamics during training, which increases the risk of training instability. (2) *biases in exploration direction.* The explorer learns solely using extrinsic rewards, while the controller uses extrinsic and intrinsic rewards. Within such a framework, the strategies of the controller and the explorer inevitably differ, and using the intrinsic rewards generated by the explorer to guide the controller can lead to biased, even wrong, exploration direction. Just as when we seek advice from others, their suggestions often do not suit us best due to differences in values or other factors.

Thereby, we propose a novel multi-agent exploration framework named Farsighted Self-Direction (FSD). The first distinction from previous curiosity-driven methods is that we forgo the explorer. The framework contains only two network components: the controller Q and the predictor Q. We use the prediction error against the controller's individual Q-values by the predictor Q to measure the potential exploration bonus. By eliminating the intermediary explorer, FSD avoids the exploration direction bias resulting from the strategy discrepancy. Furthermore, we only use extrinsic rewards to update the parameter of the controller, circumventing the instability caused by time-varying intrinsic rewards. The usage of prediction error leads to our second distinction: we do not treat prediction error as an immediate intrinsic reward. Instead, we consider it as a long-term potential exploration gain, referred to as $Q_i^{int}$. $Q_i^{int}$ is added with controller Q-values, together dominating action selection. Intuitively, as training progresses, both the controller Q and the predictor Q will gradually converge, causing the prediction error to become small enough (mainly dependent on the fluctuation of the controller Q). Thus, it can be reasonably assumed that the prediction error represents the current estimation of the upper bound of the benefits that can be obtained through exploration in subsequent training.

FSD uses the prediction error to directly bias the action selection, which requires our prediction error to accurately reflect the novelty of the state-action pair and the strength of the agents' interdependence in the current state. Similar to the approach used in the Twin Delayed Deep Deterministic Policy Gradient (TD3) (Fujimoto et al., 2018), we introduce clipped double Q-learning to mitigate the overestimation problem of the controller Q, thereby reducing the noise in $Q_i^{int}$ caused by the original Q function approximation error.

We validate our method on didactic examples used in EMC (Zheng et al., 2021). By illustrating the distribution of agents' positions at different times during training, we demonstrate that our method can effectively guide agents to explore states with greater potential value. Subsequently, we evaluated FSD on Predator Prey (Rashid et al., 2020) and several challenging StarCraft II micromanagement tasks (Samvelyan et al.,

2019). Empirical results show that FSD achieves state-of-the-art performance compared to several widely adopted baseline methods. Notably, unlike previous curiosity-driven methods, FSD does not use an explorer, thereby saving a considerable amount of computational resources. Additionally, we demonstrate the effectiveness of $Q_i^{int}$ and clipped double Q-learning separately through ablation studies.

**Contributions.** (1) We introduce FSD, a novel approach that utilizes the prediction error against the controller Q as the long-term exploration bonus to achieve farsighted exploration and forgoes the explorer Q to conserve computational resources and achieve self-directed exploration. (2) We provide an intuitive analysis of the prediction error and demonstrate the soundness of FSD along with clipped double Q-learning. (3) We show that FSD can be seamlessly combined with value decomposition approaches to enhance their performance, conducting experiments on Simultaneous Arrival, Predator Prey, and several challenging tasks in the StarCraft II micromanagement benchmark.

## 2 Background

**Decentralized Partially Observable Markov Decision Process.** In practice, the full state of the environment is often unavailable to agents, and agents can only have observations of the environment. In such cases, a cooperative multi-agent task can be modeled as a Decentralized Partially Observable Markov Decision Process (Dec-POMDP) (Oliehoek & Amato, 2016). Dec-POMDP can be described by a tuple $G = \,<\mathcal{I}, \mathcal{S}, \mathcal{A}, P, \Omega, O, R, n, \gamma>$ where $\mathcal{I}$ represents the set of $n$ agents, $\mathcal{S}$ is the global state space and $\mathcal{A}$ is the action space. At each time step, agent $i \in I$ observes $o_i \in \Omega$ determined by observation function $O(s, i)$ and its action-observation history will be updated as $\tau_i \in \tau \equiv (\Omega \times \mathcal{A})^* \times \Omega$. Then agent $i$ selects an action $a_i$ according to its policy $\pi_i$, which aims to maximize the long-term team reward jointly. After the joint action $\boldsymbol{a} \equiv [a_i]_{i=1}^n \in \boldsymbol{\mathcal{A}} \equiv \mathcal{A}^n$ has been conducted, all agents receive the feedback from the environment, which includes a shared reward $r = R(s, \boldsymbol{a})$ and new agent-specific observation $o_i' = O(s', i)$. $s'$ is the new state determined by transition probability function $P(s'|s, \boldsymbol{a})$. The objective of agents is to find the joint policy $\boldsymbol{\pi}$ that maximize the expectation of discounted cumulative team reward $\mathbb{E}[\sum_{t=0}^{\infty} \gamma^t r_t | s = s_0, \boldsymbol{\pi}]$, or simply denoted as the joint value function $V^{\boldsymbol{\pi}}(s)$, where $\gamma \in [0, 1]$ is the discount factor.

**Centralized Training with Decentralized Execution (CTDE).** In this paradigm, global information is utilized during training, guiding agents in learning superior strategies while circumventing the non-stationarity problem inherent in independent learning, thus allowing for the convenient use of replay buffer to enhance sample efficiency. During execution, agents' policies rely solely on local action-observation history. Except for decentralized actor, centralized critic approaches based on actor-critic framework (Lowe et al., 2017; Yu et al., 2022), value decomposition approaches (Sunehag et al., 2018; Rashid et al., 2018; Son et al., 2019; Wang et al., 2021b) are the mainstream methods for the CTDE paradigm, where the global Q value is decomposed into a combination of local Q-values:

$$Q_{tot}(\boldsymbol{\tau}, \boldsymbol{a}) = f(Q_1(\tau_1, a_1), ..., Q_n(\tau_n, a_n)). \tag{1}$$

$Q_{tot}$ is trained to minimize the expected TD error:

$$\mathcal{L}(\boldsymbol{\theta}) = \mathbb{E}_{\boldsymbol{\tau}, \boldsymbol{a}, r, \boldsymbol{\tau}' \in D} \left[ r + \gamma \max_{\boldsymbol{a}'} Q_{tot}^{\boldsymbol{\theta}^-}(\boldsymbol{\tau}', \cdot) - Q_{tot}^{\boldsymbol{\theta}}(\boldsymbol{\tau}, \boldsymbol{a}) \right]^2, \tag{2}$$

where $D$ represents the replay buffer and $\boldsymbol{\theta}^-$ is the parameter for the target network. $\boldsymbol{\theta}^-$ is regularly updated by $\boldsymbol{\theta}$, serving as a latent and stable follower of $\boldsymbol{\theta}$. After training, the execution policy is derived from $Q_i$.

VDN (Sunehag et al., 2018) is the earliest method based on value decomposition, which simply factorizes $Q_{tot}$ into the sum of individual Q-values. QMIX (Rashid et al., 2018) highlights that, for value decomposition approaches, it is essential to ensure the following consistency between $Q_{tot}$ and individual $Q_i$:

$$\arg\max_{\boldsymbol{a}} Q_{tot}(\boldsymbol{\tau}, \boldsymbol{a}) = \left\{ \begin{array}{c} \arg\max_{a_1} Q_1(\tau_1, a_1) \\ ... \\ \arg\max_{a_n} Q_n(\tau_t, a_n) \end{array} \right\}, \tag{3}$$

which is later denoted as the IGM principle (Son et al., 2019). Consequently, QMIX introduces a hypernetwork $f$ as eq. 1 and ensures that $\frac{\partial f}{\partial Q_i} \geq 0, \forall i \in \mathcal{I}$. As a classic method, QMIX usually provides stable and

satisfactory performance. Subsequently, Weighted-QMIX (Rashid et al., 2020) introduces weighting into Q-value learning and places more importance on better joint actions to address the limitation of monotonicity. More algorithms have been proposed to enhance representation expressiveness, such as QTRAN (Son et al., 2019), QPLEX (Wang et al., 2021b) and DAVE (Xu et al., 2023).

## 3    Related Work

**Single-agent exploration.**    In the single-agent domain, how to achieve effective exploration has been extensively studied. Recent literature (Aubret et al., 2019; Hao et al., 2024) has already provided a comprehensive summary of the topic, especially on intrinsic motivation-oriented methods, also known as curiosity-driven methods (Zheng et al., 2021). Intuitively, the novelty of certain states can be measured by the visit times to those states, and intrinsic rewards can be calculated accordingly. To extend these methods to large state space, various methods with pseudo-count have been introduced (Bellemare et al., 2016; Fox et al., 2018) where DORA (Fox et al., 2018) suggests leveraging the pseudo-count as the long-term exploratory value. RND (Burda et al., 2019), on the other hand, assesses state novelty through the prediction error of the next state embedding, which is obtained from a randomly initialized, fixed network and predicted by another network with the same structure but learnable. Moreover, VIMIM (Mohamed & Rezende, 2015) encourages agents to explore states with higher empowerment, calculated using the mutual information between action sequences and states. VIME (Houthooft et al., 2016) also utilizes information theory, encouraging agents to maximize information gain about the environment dynamics with each step, thus satisfying the agent's curiosity about the environment. ICM (Pathak et al., 2017) uses an inverse model for better state embedding, allowing the forward model to be less affected by the "noisy TV" problem (Burda et al., 2019) when predicting the next state. Recently, NGU (Badia et al., 2020) leverages both episode and inter-episode intrinsic rewards, achieving better performance in long-sequence problems.

**Multi-agent exploration.**    Unlike exploration in the single-agent setting, efficient coordinated exploration is crucial in multi-agent environment. Jaques et al. (2019) proposes calculating intrinsic rewards for agents' chosen action by assessing its influence on other agents' action selection. Subsequently, EITI and EDTI (Wang et al., 2020) incorporate agents' interaction values as regularization terms in policy optimization, where EITI encourages exploration of states that influence other agents' transition dynamics while EDTI focuses on states that affect other agents' decision-making. IIE (Liu et al., 2024) leverages transformer models to identify interaction states and guides agents to start exploration from these high-influence states. MACE (Jiang et al., 2024) is a method under the distributed learning setting with limited communication, facilitating coordinated exploration by sharing local novelty and hindsight influence estimation.

On the other hand, MAVEN (Mahajan et al., 2019) introduces a latent variable to learn hierarchical policies, enabling committed exploration through diverse exploration strategies. CDS (Li et al., 2021) encourages agents to exhibit diverse exploratory behaviors by incorporating mutual information between agents as a regularization term. EMAX (Schäfer et al., 2024) explicitly uses ensemble value functions to estimate the variance among Q-values and follows the UCB (Auer et al., 2002) exploration policy.

EMC (Zheng et al., 2021) is the first to use the prediction error of individual Q-values as intrinsic rewards. Since value decomposition methods achieve implicit credit assignment Wang et al. (2021a), the prediction error of the individual Q-value function is also influenced by other agents; thereby, agents are encouraged to visit states where careful coordination is required. Building on this, EXPODE (Zhang & Yu, 2023) uses twin QMIX and intricately designs the intrinsic reward as the prediction error against the higher Q-value. However, the controller-explorer-predictor structure used in both EMC and EXPODE inevitably leads to deviation in the controller's exploration direction.

## 4    Method

This section introduces FSD, a novel model-free method for efficient multi-agent exploration. FSD takes the prediction error against individual controller Q-values as $Q_i^{int}$, which is combined with controller Q-values to determine action selection during sample collection. First, we describe the method in detail; then, we

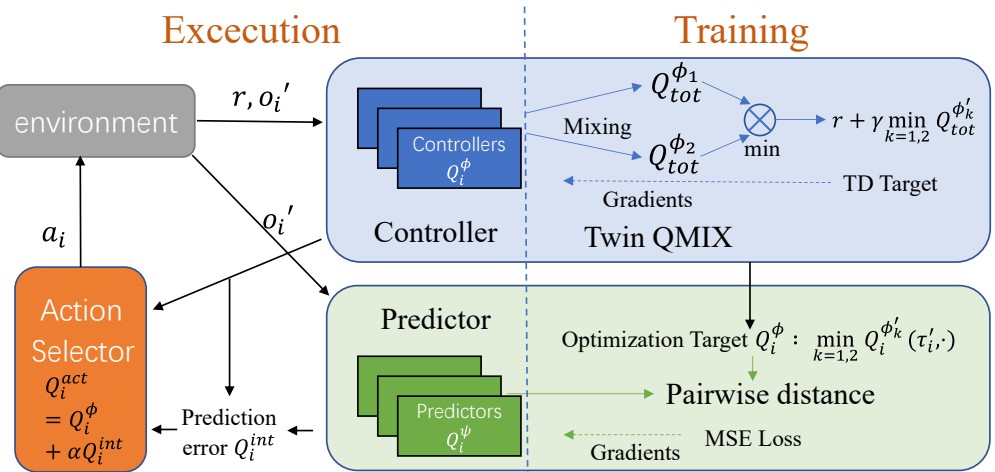

Figure 1: An overview of FSD's framework

analyze the sources of prediction error to verify the significance of $Q_i^{int}$ and why clipped double Q-learning can help reduce noise in prediction error.

### 4.1 Farsighted Self-Direction

The prediction error against individual Q-values contains not only information about local exploration but also global exploration status due to implicit credit assignment (Zheng et al., 2021; Wang et al., 2021a). Thus, we use the prediction error against individual Q-values as the signal to guide agents in coordinated exploration. As shown in Fig. 1, FSD contains two network components, namely the controller and the predictor, where the controller represents the strategy of agent interaction with the environment, and the predictor is trained to approximate the controller. The prediction error between the predictor and the controller serves as the signal for directing coordinated exploration.

Additionally, FSD considers the prediction error as potential long-term exploration gain $Q_i^{int}$ in a farsighted manner rather than immediate reward gained from exploration. Over the long term, the controller and the predictor gradually converge after sufficient exploration, and the prediction error will eventually become a value small enough. Thus, to some extent, the prediction error represents the long-term exploration bonus that can be obtained from exploration starting from the current state-action pair. In the action selection phase, FSD calculates $Q_i^{int}(\tau_i, \cdot)$, which, along with the controller $Q_i^\phi(\tau_i, \cdot)$, guides the action selection process:

$$Q_i^{act}(\tau_i, \cdot) = Q_i^\phi(\tau_i, \cdot) + \alpha Q_i^{int}(\tau_i, \cdot), \tag{4}$$

where $\alpha$ is used to control the magnitude of exploration. Subsequently, $\epsilon$-greedy strategy based on $Q_i^{act}$ determines the final selected action.

Let the predictor be denoted by $Q_i^\psi$, $Q_i^{int}$ is calculated as the distance between the controller and the predictor:

$$Q_i^{int}(\tau, \cdot) = \left| Q_i^\phi(\tau_i, \cdot) - Q_i^\psi(\tau_i, \cdot) \right|, \tag{5}$$

As analyzed in section 4.2, larger $Q_i^{int}(\tau_i, a_i)$ with specific $a_i$ indicates more insufficient visits to the state-action pair or stronger interdependence among agents existing in the next state after $a_i$ is taken, which makes the action more worth taking and the next state more worth exploring. With $Q_i^{int}$ biasing agents into coordinated exploration, more valuable experience is collected, and agents are likely to achieve a better policy.

QMIX (Rashid et al., 2018), a classic method performing well across a multitude of tasks, is selected as the backbone. It's worth noting that our method can be embedded into any value decomposition method.

As shown in eq. 5, we directly use the pointwise prediction error, which requires that our prediction error accurately reflects the information for coordinated exploration. The inherent over-estimation problem in Q-learning can lead to fluctuations in the controller Q-values, which constitute a significant source of noise in the prediction error. To reduce noise, FSD introduces clipped double Q-learning similar to TD3 (Fujimoto et al., 2018), mitigating the over-estimation problem by taking the smaller target Q to form the TD target.

Specifically, the controller includes two networks with the same structure but different initializations, $Q_i^{\phi_1}$ and $Q_i^{\phi_2}$, where $Q_i^{\phi_1}$ learns the policy to interact with the environment (referred to as $Q_i^{\phi}$ in eq. 4 and eq. 5) while $Q_i^{\phi_2}$ is only used during training. Correspondingly, two target networks, $Q^{\phi_1'}$ and $Q^{\phi_2'}$, are used to stabilize training, and the smaller target Q-value is used to calculate the one-step TD target:

$$a_k' = \arg\max_a Q_{tot}^{\phi_k}(\tau', \cdot), \qquad k = 1, 2 \tag{6}$$

$$\mathcal{L}_{controller}(\phi_k) = \left[ r + \gamma \min_{k'=1,2} Q_{tot}^{\phi_{k'}'}(\tau', a_{k'}') - Q_{tot}^{\phi_k}(\tau, a) \right]^2, \qquad k = 1, 2 \tag{7}$$

$$\phi_k' \leftarrow (1 - \xi)\phi_k' + \xi\phi_k, \qquad k = 1, 2 \tag{8}$$

The predictor has the same network structure with the local $Q_i^{\phi}$ and is trained concurrently with the controller to approach it:

$$\mathcal{L}_{predictor}(\psi) = \frac{1}{n} \sum_{i=1}^{n} \left\| Q_i^{\psi}(\tau_i', \cdot) - \min_{k'=1,2} Q_i^{\phi_{k'}'}(\tau_i', \cdot) \right\|_2. \tag{9}$$

### 4.2 Reduce Noise in Prediction Error

The prediction error of the individual controller Q-values stems from two main sources:

1. *Moving target.* As the optimization target for the predictor, the individual controller Q-values are continually changing along with the advancement of the training, which makes the target difficult to approach.

2. *Error in approaching a fixed target.* Even if the individual controller Q-values were fixed, there would still be an approximation error between the predictor and the fixed target.

Since both the predictor and controller share the same local $Q_i$ network structure, the error in approaching a fixed target primarily depends on how frequently a particular state-action pair (or its neighboring spaces) is visited, which indicates the novelty of the state-action pair $(s, a)$ and usually the novelty of the next state $s'$.

The fluctuations in the individual controller Q-values can be further broken down into three major components:

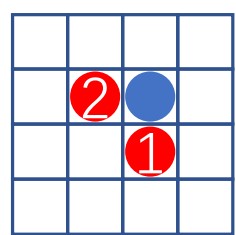

1. *Infrequent visits to a particular state-action pair.* Few visits to some state-action pair make the estimation of its value unstable.

2. *Strong interdependence among agents in some states.* If strong interdependence lies among agents in some states, the estimation of individual Q-values will be highly susceptible to the actions chosen by other agents during their exploration.

3. *Overestimation in Q-learning.* It has been known that Q-value is easily overestimated in training, and such positive bias becomes more pronounced as error accumulates (Liang et al., 2022).

Figure 2: Strong interdependence lies between agents in the state.

To better illustrate the second component, let's consider the scenario shown in Fig. 2: two agents in red are required to arrive at the blue target simultaneously, and any individual arrival would be punished. If Agent 1 moves up, the reward may be positive if Agent 2 moves right, or negative otherwise. On the other hand, if Agent 1 takes other actions, the reward may be negative if Agent 2 moves right, or zero otherwise. Compared to the other actions, the move-up action of Agent 1 in the current state is more influenced by the other agent. And the fluctuation of the controller $Q_1(\tau_1, a_{up})$ is likely larger, which leads to a higher $Q_1^{int}(\tau_1, a_{up})$, guiding the agent to actively try to reach the target. Besides, $Q_i^{int}$ also guides the agents toward similar states with strong agent interdependence.

We aim for agents to explore novel states or interaction states where strong interdependence lies. From the analysis above, we show that among the listed factors contributing to the prediction error of individual controller Q-values, only the fluctuations caused by Q-value over-estimation constitute noise. Therefore, akin to TD3, we employ clipped double Q-learning to significantly mitigate the problem of Q-value overestimation, thereby effectively reducing noise in prediction error.

EXPODE (Zhang & Yu, 2023) also uses twin QMIX, however, EXPODE primarily aims to leverage the min-max operator to construct better intrinsic rewards, with the max operator employed in generating intrinsic rewards. In contrast, our approach solely focuses on reducing noise in prediction error, and we simply use the prediction error between the predictor Q and the smaller target controller Q-value, which is consistent with the update of controller Q in TD target formation.

## 5 Results

In this section, we conduct a set of empirical experiments and demonstrate the performance of FSD in Simultaneous Arrival (Zheng et al., 2021), Predator Prey (Rashid et al., 2020), and StarCraft II micromanagement (Samvelyan et al., 2019). QMIX is the default backbone if not specified. Experiments are conducted on eight NVIDIA RTX 4090s, with training time ranging from half an hour to 10 hours, depending on the complexity of the task and the number of agents involved. Baselines are trained using their open-source codes, with some results derived from the open-source results of EXPODE (Zhang & Yu, 2023). As for evaluation, action is selected greedily based on the controller $Q_i^{\phi}$. All experiments have been repeated for five runs over different random seeds. In the ablation study section, we separately showcase the effectiveness of $Q_i^{int}$ and clipped double Q-learning.

### 5.1 Simultaneous Arrival

Simultaneous Arrival is introduced by Zheng et al. (2021) as a didactic example. In this task, two agents are separated from the middle, and they can only observe each other when they are near their corresponding targets. A positive reward is given only if both agents arrive at their corresponding targets simultaneously, while a negative reward $-p$ is given if only one agent reaches its target. Otherwise, agents receive no reward. In our experiments, we set $p$ to a relatively high value of 3. The success of the task requires a high level of coordinated exploration from the agents, and its grid-world environment also facilitates visualization.

It has been validated that EMC-VDN performs better than its counterparts in this task, so our comparison is conducted among methods using VDN as the backbone. The tested methods include FSD-VDN, FSD-sgl-VDN denoting no clipped double Q-learning is used, FSD-woin-VDN denoting no $Q_i^{int}$ is used, EXPODE-VDN, EMC-VDN, and VDN.

As illustrated in Fig. 3, both FSD-VDN and EMC-VDN achieve success in this task. However, EXPODE-VDN and FSD-sgl-VDN frequently fail in this task despite occasional successes. Similar to VDN, FSD-woin-VDN consistently fails to complete the task. These results affirm the necessity of $Q_i^{int}$ for FSD in the Simultaneous Arrival task. Notably, the standard setting of the scenario lacks randomness, meaning that during evaluation, the win rate is either 0 or 1 for a fixed policy, and the curves essentially represent the proportion of wins across five different runs.

Besides, as shown in Fig. 4, we have plotted the heat map of the frequency of visitation to various locations by agents at different times during the training of FSD-VDN. In the very early stages of training, the agents

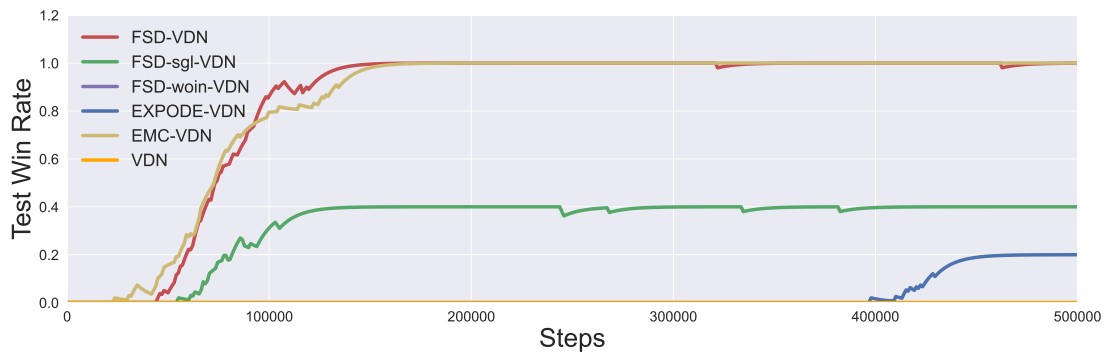

Figure 3: The performance on Simultaneous Arrival

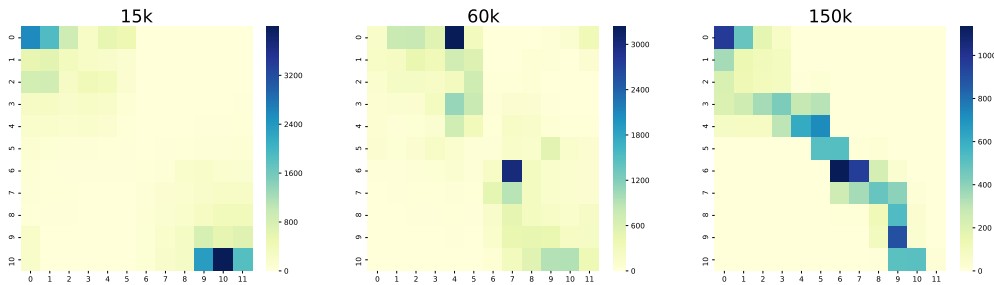

Figure 4: The visitation frequency heat map on Simultaneous Arrival during the training of FSD-VDN

broadly explore their surrounding environment, while in this case, each agent primarily moves around the starting point, the upper left and lower right corners of the map, respectively. Directed by $Q_i^{int}$, agents gradually move closer to the center of the map, where the targets are located. By the 150k time steps, the agents are almost always successful in completing the task, reaching the targets on row 5 simultaneously. The heat map shows that the agent on the right side visits the area near its target more frequently than the agent on the left, indicating that it has learned a strategy to wait for the other agent before reaching the target.

## 5.2 Predator Prey

Predator Prey is a challenging, partially observable multi-agent game where predators must cooperate to capture prey and receive rewards, with penalties imposed for individual captures. This scenario features a larger number of agents (eight) and includes mobile prey, making it more challenging than the simultaneous arrival task. We conduct comparisons of the FSD against several baseline algorithms, including EXPODE, EMC, Weight-QMIX (Rashid et al., 2020), QMIX, and VDN.

As shown in Fig. 5, FSD-VDN and FSD-QMIX outperform other baselines in learning efficiency. EXPODE-VDN and EMC-VDN also succeed in the task and demonstrate strong performance. It is worth noting that, in contrast to FSD-QMIX, EXPODE-QMIX fails in the task and tends toward a local optimum where predators do not capture prey at all, which indicates our method can

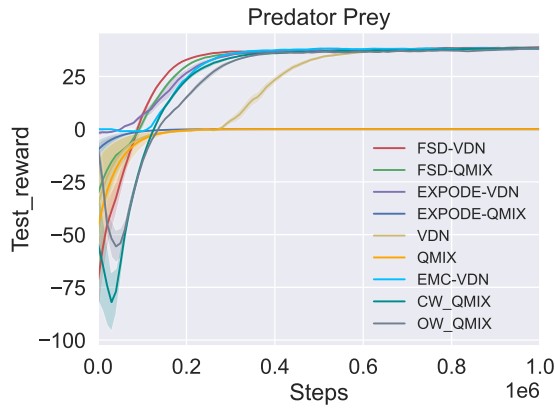

Figure 5: The performance on Predator Prey

better enhance coordinated exploration in this task. Among other baseline methods, Weighted-QMIX is noted for its ability to focus on learning the value of $Q(s, a^*)$ where $a^* = \arg\max_a Q(s, a)$, thus also finds the optimal policy. VDN also succeeds in the task, albeit much less efficiently. QMIX, which is based on the monotonicity assumption, fails in the task due to its non-monotonic reward structure.

### 5.3 SMAC Benchmark

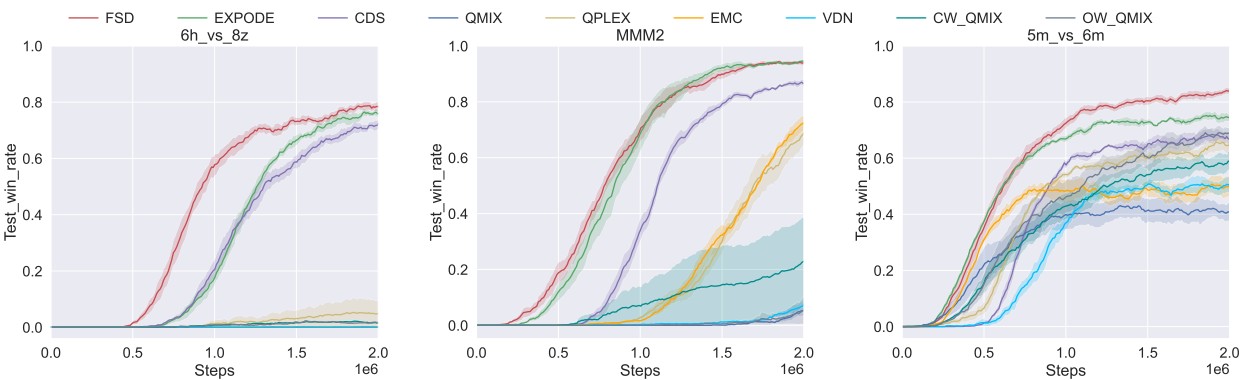

Figure 6: Results on SMAC benchmark

StarCraft II Micromanagement (SMAC) is a commonly used benchmark in the field of multi-agent reinforcement learning (Wang et al., 2021b; Zheng et al., 2021; Zhang & Yu, 2023). In SMAC, agents trained by the evaluated methods have only partial observation of the environment, while their opponents are driven by the built-in AI. We tested FSD on two super hard maps: 6h_vs_8z and MMM2, and one hard map: 5m_vs_6m. Baseline algorithms include EXPODE, EMC, CDS, QMIX, QPLEX, VDN and Weighted-QMIX. As illustrated in Fig. 6, FSD has achieved state-of-the-art performance.

CDS fosters better exploration by encouraging diversity among agents. EMC, with the compact expressive capabilities of QPLEX, utilizes intrinsic rewards and episodic memory to direct coordinated exploration and make the best use of experience, demonstrating strong performance in the SMAC benchmark. EXPODE further improves performance through the design of intrinsic rewards and enhanced learning of the predictor network.

In 6h_vs_8z, FSD exhibits the fastest learning rate, and in 5m_vs_6m, it achieves the highest win rate, while in MMM2, FSD achieves performance indistinguishable that of EXPODE. FSD leverages the prediction error against the controller Q to directly formalize $Q_i^{int}$ and reduce the noise in $Q_i^{int}$ through the implementation of clipped double Q-learning, thereby more accurately directing agents in coordinated exploration.

### 5.4 Ablation Study

**Effect of $Q_i^{int}$ and clipped double Q-learning.** We conduct ablation studies on Predator Prey and the super hard 6h_vs_8z and MMM2 maps of SMAC to specifically assess the impacts of $Q_i^{int}$ and clipped double Q-learning. Note that removing these two components reduces our algorithm to QMIX. FSD without $Q_i^{int}$ is denoted as FSD-wo-in, and FSD without clipped double Q-learning is denoted as FSD-sgl.

Fig. 7(a) shows that on Predator Prey, FSD-wo-in does not show improvement over QMIX and similarly fails on the task. In contrast, FSD-sgl succeeds in the task and achieves indistinguishable performance from FSD. This suggests that, in relatively simpler scenarios where the overestimation issue is less severe and also less noise in $Q_i^{int}$, clipped double Q-learning does not have a significant impact. On the other hand, $Q_i^{int}$ can help find the optimal policy by directly enhancing coordinated exploration.

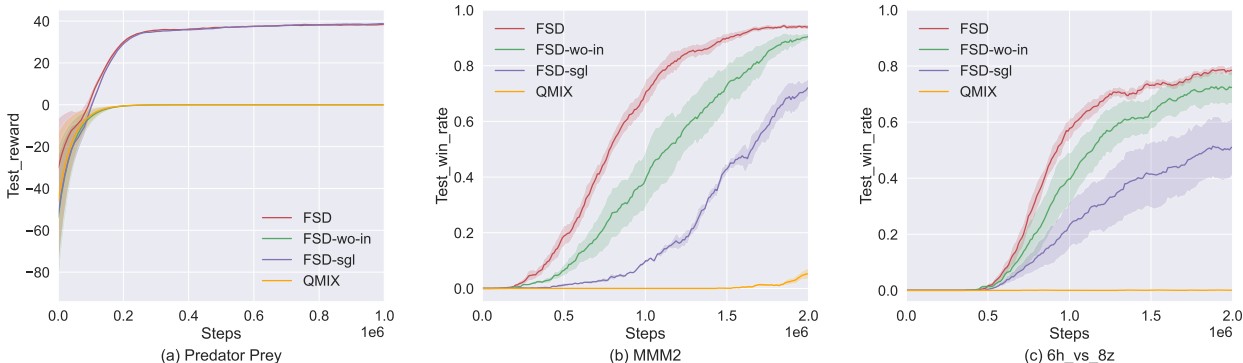

Figure 7: Ablation study on the two major components

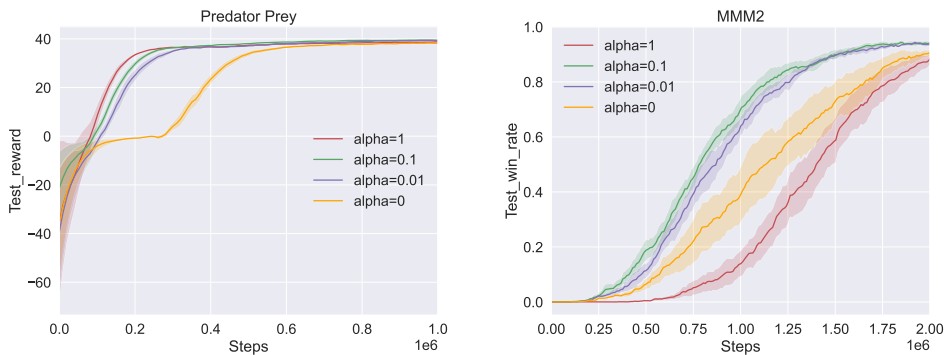

Figure 8: Ablation study on the coefficient term $\alpha$

It can be observed from Fig. 7(b),(c) that clipped double Q-learning is crucial for agents to learn effective strategies on these two maps due to the complex environments where over-estimation issues can destabilize training and introduce much noise in $Q_i^{int}$. $Q_i^{int}$ also plays an important role, as FSD-sgl shows performance improvements over QMIX on both maps. Additionally, with higher learning efficiency, FSD achieves better and more stable performance compared to FSD-wo-in.

Across both the complex SMAC tasks and the simpler Predator Prey task, combining clipped double Q-learning with $Q_i^{int}$ maximizes the effect of $Q_i^{int}$, more accurately directing agents in coordinated exploration and achieving high performance stably.

**Effect of exploration coefficient** $\alpha$. We also analyze the effect of the exploration coefficient $\alpha$ in eq. 4 on performance using FSD-sgl-VDN and FSD in the Predator Prey and the MMM2 map of SMAC, respectively. Fig. 8 shows that, in general, a coefficient between 0.01 and 0.1 can effectively improve exploration efficiency. In the relatively simple Predator-Prey scenario, where coordinated exploration is crucial, $\alpha$ can take a larger value, such as 1, to further enhance exploration. However, on the more complex MMM2 map, setting $\alpha = 1$ leads to excessive exploration by the agents, which ultimately reduces their learning efficiency.

## 6 Conclusion

This paper introduces FSD, a simple yet effective method for enhancing efficient multi-agent exploration through farsighted self-direction. FSD leverages a long-term exploration bonus, tailored for the agent, as a directional signal, which is denoised using clipped double Q-learning. With this efficient exploration method, our approach has outperformed a wide range of widely adopted baseline methods on the challenging SMAC

II benchmark. The limitation of our work lies in the lack of robustness against fluctuation in prediction error. For future work, we may consider using methods that adaptively adjust exploration magnitude to enhance robustness. Additionally, we will evaluate our method on other influential benchmarks or different application scenarios.

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
