# OpenReview forum: "Efficient Exploration in Multi-Agent Reinforcement Learning via Farsighted Self-Direction"
_TMLR — Accepted by TMLR_

### Review · Reviewer_uMaH · 2025-03-01

**Summary Of Contributions:**

The paper proposes a novel method for multi-agent coordinated exploration based a simpler way of computing the intrinsic-motivation reward to guide it. This farsighted self-directed motivation removes the need for an explorer in the algorithm, instead only leaving the exploitation behaviours and the predictor one, that is used to compute the error with the former that acts as the exploration incentive. The algorithm uses double-clipped $Q$-learning as a building component, and can be combined with different value-decomposition methods.

**Audience:**

Yes

**Claims And Evidence:**

Yes

**Requested Changes:**

- In the Introduction, you say that in intrinsic-motivated exploration, the exploiter is used to interact with the environment, while the explorer is used to provide the intrinsic reward (and thus the exploration strategy). Why then you say a bit later than the explorer learns from the extrinsic (real) reward, while the exploiter learns from both? Shouldn't it be the other way around? Perhaps I misunderstood how this is supposed to work? Moreover, what does it mean that the explorer uses a different predictor? That is is using a different policy? How does this come into play? In any case, I suggest you carefully rephrase this paragraph to be clearer and more explicit, especially as it is then useful to understand the differences with your proposed method, underpinned in the paragraph below.

- In the green box in Figure 1, how can you compute $\min_{k=1,2}Q_{tot}^{\phi'_k}$ on local histories $\tau'_i$? The pairwise distance is between $Q^{\phi}_i$ and the corresponding $Q^{\psi}_i$ from Equation (5), so what have the joint target networks to $Q^{\phi'_k}\_{tot}$ do here? Perhaps these should be be $Q^{\phi'_k}_i$ (as it seems from Equation (9) indeed)?

- The paragraph before Equation (6) is a bit confusing: what is double here? Only the mixed (joint) $Q$-function $Q_{tot}^{\phi}$ or also the agents' local $Q$-networks $Q_i^{\phi}$? From Equation (9) it seems that also the agents' local $Q$-networks are double, but this is not clear neither from the text nor from Figure 1 (where indeed there is a single set of local $Q$-networks in the blue box, but clearly two mixing processes coming from them, leading to two different joint $Q$-functions). I suggest you clarify these aspects a bit more in details, and adjust notations and representations everyone to be consistent with that.

- In the Simultaneous Arrival experiment, FSD-woin-VDN basically results in clipped double VDN no?

- Figure 5 is really difficult to parse correctly, and differentiate between lines with similar colours (like FSD-sgl-VDN and QPLEX for example) is hard, especially because many of them are behaving similarly and overlap quite often. I suggest to change the colour scheme for the lines to make them more easily identifiable, and perhaps to remove the less interesting algorithms to its improve readability.

- The fact that (FSD-/EXPODE-)QMIX behaves worse than its VDN variants in Predator Prey is interesting: usually QMIX has an higher expressive power than VDN (additivity is indeed a special case of monotonicity used in QMIX's constraint), but it seems to not be able to learn at all in this setting. Do you have any insight on why this may be the case (especially more so, why your more complex FSD-QMIX or EXPODE-QMIX are outperformed by a much simpler standard VDN)?

**Strengths And Weaknesses:**

The problem of achieving improved and coordinated exploration in MARL is an important one, and worth attention. The proposed idea is quite simple and general, and I like the fact that it lifts the need for an explicit explorer in the intrinsic-motivation framework. The methodology seems sound and well motivated, although the main point on the method convergence is only provided as an intuition, rather than being formally proved. The experimental results show indeed the good performance of the proposed framework, although their interpretation is not always as clear as it should, and the results analysis is a bit poor. In general, the paper is quite easy to read and follow, but notation is not always extremely clear, and this renders the understanding of some aspects a bit difficult sometimes.

---

> ### Author Response · Authors · 2025-03-24
> **Response to reviewer uMaH**
>
> We sincerely thank you for carefully reviewing our manuscript and providing detailed and constructive feedback. Our response to the requested changes can be found below and we have revised the manuscript accordingly, marking the changes in red.
>
> - rephrase the paragraph that introduces curiosity-driven methods in the Introduction section
>
>   Apologies for any misunderstanding caused by our description. Perhaps the term "exploiter" was not the most appropriate choice here. We will replace it with "controller" throughout the manuscript. **The controller interacts with the environment, collects samples (thus also incorporating the exploration policy), and serves as the optimized policy. The explorer, on the other hand, learns solely from external rewards and is used exclusively to generate intrinsic rewards to enhance exploration (hence the name "explorer"). The controller receives both intrinsic and external rewards, enabling it to be guided toward states that require further exploration.** This design separates the generation and application of intrinsic rewards, preventing excessive coupling between the two. Additionally, there is only one predictor, and the prediction error between the predictor and the explorer is used as the intrinsic reward.
>
>   **We revise this paragraph as follows:**
>
>   To address the limitation, the intrinsic reward is designed as the prediction error of the individual Q-values for value decomposition methods, promoting efficient coordinated exploration. To avoid excessive coupling between intrinsic reward generation and policy learning, these curiosity-driven methods typically employ a controller-explorer-predictor structure where the controller learns the policy used to interact with the environment and the explorer is only used during training to generate the intrinsic reward. The prediction error between the explorer and the preditor is used as the intrinsic reward for the controller to enhance exploration...
>
> - In Figure 1,  perhaps  $\min_{k=1,2}Q_{tot}^{\phi_k'}(\tau_i',\cdot)$ should be $\min_{k=1,2}Q_{i}^{\phi_k'}(\tau_i',\cdot)$
>
>   **Yes,  $\min_{k=1,2}Q_{tot}^{\phi_k'}(\tau_i',\cdot)$ should be revised to $\min_{k=1,2}Q_{i}^{\phi_k'}(\tau_i',\cdot)$.** Thank you for pointing out the mistake and we have fixed it.
>
> - The paragraph before Equation (6)  is a bit confusing: what is double here?
>
>   Apologies for the confusion here. **The agents' local Q-networks $Q_i^\phi$ is also double.** We have added the subscript $i$ to $Q^{\phi_1}$ and $Q^{\phi_2}$ in the text to make it more clear.
>
> - In the Simultaneous Arrival experiment, FSD-woin-VDN basically results in clipped double VDN no?
>
>   Yes, FSD-woin-VDN basically results in clipped double VDN.
>
> - Figure 5 is really difficult to parse correctly. It's suggested to change the color scheme and perhaps to remove the less interesting algorithms
>
>   Thank you for the suggestion. **We have changed the color scheme, removed 2 less interesting algorithms, and redrawn the figure. Figure 5 has been updated in the new manuscript.**
>
> - why QMIX and even FSD-QMIX, EXPODE-QMIX are outperformed by VDN?
>
>   QMIX has indeed a stronger representational capacity than VDN. However, QMIX can only represent the value function that satisfied the Monotonicity condition, while the Predator-Prey (PP) task exhibits strong non-monotonic characteristics. **Therefore, both QMIX and VDN may fail on the PP task.**
>
>   Returning to the experiment itself, it is indeed an interesting phenomenon that VDN succeeds while QMIX fails on this task. Theoretically, the solution space of VDN is strictly contained within that of QMIX. Although the true solution lies outside QMIX’s solution space, QMIX still has the potential to find an optimized result closer to the true solution. My conjecture is that the **greater complexity of QMIX’s network structure increases the optimization difficulty**. Due to the **simple grid environment and sparse rewards**, QMIX may easily fall into a **local optimum** (predator staying away from prey). In contrast, the **additive constraint of VDN acts as a form of regularization**, which helps it succeed in this task.
>
>   As for FSD-QMIX, **we have finetuned the parameter $\alpha$  in equation (4) and reset it to 1, it turns out that FSD-QMIX can significantly outperform VDN.  The results have been updated on the new Figure 5.** FSD-QMIX guides agents into efficient coordinated exploration, thereby acquiring more valuable experiences (such as successful experiences in the PP task). The increased proportion of successful experiences in the training samples makes QMIX more inclined to fit the optimal $a^*$. In essence, this achieves an effect similar to Weighted QMIX.
>
>   EXPODE-QMIX is able to improve the exploration, but the improvement here is not enough to succeed in the task.

---

### Review · Reviewer_z7h1 · 2025-03-03

**Summary Of Contributions:**

The paper introduces a novel method, Farsighted Self-Direction (FSD), to promote exploration in MARL. The proposed approach utilizes prediction error as a long-term exploration bonus, facilitating self-directed exploration. When integrated with value decomposition methods, FSD enhances performance.

**Audience:**

Yes

**Broader Impact Concerns:**

N/A.

**Claims And Evidence:**

No

**Requested Changes:**

Critical Issues:

1. Provide a more thorough analysis of why and how the proposed method works, specifically how prediction error is captured by $Q_i^{int}$ and how this contributes to improved performance.

2. Include a more detailed comparison with EXPODE, clarifying the motivation behind FSD and the specific limitations of EXPODE that it addresses.

Additional Suggestions:

1. Address the other weaknesses outlined above.

2. Strengthen the theoretical and empirical justifications for the proposed method.

**Strengths And Weaknesses:**

**Strengths**:

- The paper provides a comprehensive background on the topic.

- The idea of using prediction error to encourage exploration is interesting.

- Illustrative examples are included to aid in understanding the method.

- Recent baselines are incorporated in the comparison, ensuring a fair evaluation.

**Weaknesses**:

1. Unclear Method Introduction

The concept of "prediction error," which is central to the proposed approach, is not formally defined.
The definition of $Q_i^{int}$ should be introduced earlier, given its frequent usage. The phrase "a long-term potential exploration gain" is vague. The paper should explicitly define $Q_i^{int}$, for example, as the expected return of intrinsic rewards.

2. Lack of Clarity in Justification

It is unclear why the proposed method works. The explanation in Section 4.2 is not fully consistent with the actual method.
The paper categorizes errors of individual controllers as "moving target" errors and errors in "approaching a fixed target," but it is not evident how $Q_i^{int}$ captures these errors.
Section 4.2 states that greater fluctuation in the controller leads to a higher $Q_i^{int}$, thereby promoting exploration. However, this is not apparent from Equations (5), (7), and (9). The paper should provide additional evidence on how reward fluctuations influence the controller, the predictor, and ultimately $Q_i^{int}$.

3. Lack of Detailed Comparison with EXPODE

The proposed method bears strong similarities to EXPODE, but the paper does not provide a clear distinction between the two.
What specific limitations of EXPODE does FSD aim to address?
Is there a strong motivation that justifies the development of FSD instead of EXPODE?

4. Limited Relevance to MARL

The proposed method does not appear to be specifically tailored for MARL.
While it can be integrated into value factorization methods, the method itself lacks specialized design elements for MARL.

5. Unclear Evidence on Exploration Efficiency

The performance improvements may primarily stem from clipped double Q-learning rather than the proposed exploration method.
The experiments do not sufficiently demonstrate that the efficiency gains result from improved exploration.
The paper could include additional experiments to analyze how the coefficient $\alpha$ influences performance and whether $Q_i^{int}$ is actually larger in less-visited states with fluctuating values.

---

> ### Author Response · Authors · 2025-03-24
> **Response to reviewer z7h1 (1/3)**
>
> We appreciate your time in reviewing our paper and your helpful comments. Please find our response below and we have revised the manuscript, marking the changes in red.
>
> Before we address the critical issues, we would like to first respond to the weaknesses.
>
> - W1:Unclear Method Introduction
>
>   We sincerely apologize for any confusion caused by our description.
>
>   **Prediction error**.The term prediction error here carries its literal meaning—**it exactly refers to the error in predicting something.** The phrase prediction error is widely used, and many methods leverage the prediction error of something (e.g. state-embedding) to capture its novelty (e.g., RND, ICM, EMC, EXPODE). Specifically, in our case, we use the predictor to predict the individual controller Q-values, so the prediction error here refers to the error for the predictor in predicting the controller. As stated in Equation 5, **$Q_i^{int}$ serves as a metric for this prediction error.**
>
>   **$Q_i^{int}$**. We have introduced $Q_i^{int}$ the first time we mention it. As mentioned in the Introduction, "we do not treat the prediction error as an immediate intrinsic reward. Instead, we consider it a long-term potential exploration gain, which we refer to as $Q_i^{int}$." **On one hand, we interpret the prediction error as a long-term gain; on the other hand, we add it to the individual controller Q-values to obtain $Q_i^{act}$. Thus, we denote it as $Q_i^{int}$.**
>
>   **It is important to clarify that $Q_i^{int}$ defined in equation (5) is our own proposed notation, not an existing standard concept.** It is not strictly defined as the expected return of intrinsic rewards. Instead, we provide an intuitive explanation of why the prediction error can be interpreted as a long-term exploration gain:
>
>   "As training progresses, the predictor gradually converges, causing the prediction error to become small enough (mainly dependent on the fluctuation of the controller Q-value). Thus, it can be reasonably assumed that the prediction error represents the current estimation of the upper bound of the benefits that can be obtained through exploration in subsequent training."
>
> - W2：Lack of Clarity in Justification
>
>   We apologize for any lack of clarity in our description. **Here, we provide additional clarifications and explanations for Section 4.2.** First, as defined in Equation (5), $Q_i^{int}$ is a measure of the prediction error between the predictor and the individual controller Q-values. Therefore, analyzing the prediction error is essentially analyzing $Q_i^{int}$. Second, as seen from Equation (7), fluctuations in $r$ or the Q-values of the next state result in fluctuations in $Q_{tot}^ {\phi_k}$ and also $Q_i^{\phi_k}$(i.e., moving target), which makes the $Q_i^{\phi_k}$ difficult to approach. Consequently, this leads to larger prediction error and thus larger $Q_i^{int}$.
>
> - W3：Lack of Detailed Comparison with EXPODE
>
>   In fact, we have already  **explained the differences between our approach and previous curiosity-driven methods (including EXPODE) in the introduction section（paragraph 3 and paragraph 4). To improve clarity, we have rephrased paragraph 3.** We hope this addresses your concern。
>
>   **We revise this paragraph as follows:**
>
>   To address the limitation, the intrinsic reward is designed as the prediction error of the individual Q-values for value decomposition methods, promoting efficient coordinated exploration. To avoid excessive coupling between intrinsic reward generation and policy learning, these curiosity-driven methods typically employ a controller-explorer-predictor structure where the controller learns the policy used to interact with the environment and the explorer is only used during training to generate the intrinsic reward. The prediction error between the explorer and the preditor is used as the intrinsic reward for the controller to enhance exploration...
>
> - W4：Limited Relevance to MARL
>
>   **The proposed method is specifically designed for MARL as we choose the individual Q-values as the prediction target.** And the prediction error serves as the exploration signal. As highlighted in EMC and further elaborated in Section 4.2, the prediction error against the individual Q-values not only captures the novelty of state-action pairs but also reflects the interdependencies among agents. This design is unique to the multi-agent setting. In contrast, in the single-agent domain, there is no need for "coordinated" exploration. Instead, the embedding of the next state is typically used as the prediction target (e.g., RND, ICM).

---

> ### Author Response · Authors · 2025-03-24
> **Response to reviewer z7h1 (2/3)**
>
> - W5：Unclear Evidence on Exploration Efficiency
>
>   Thank you for your suggestion and we have added more experiment results. First, we add new abalation study results on Predator Prey. On Predator Prey, FSD-wo-in does not show any improvement over QMIX and similarly fails on the Predator-Prey (PP) task. In contrast, FSD-sgl succeeds in the PP task and achieves indistinguishable performance from FSD. This suggests that in relatively simpler scenarios where the overestimation issue is less severe and less noise in $Q_i^{int}$, clipped double Q-learning does not have a significant impact. On the other hand, **$Q_i^{int}$** can help find the optimal policy by directly enhancing coordinated exploration. As for hard tasks like  MMM2 and 6h_vs_8z,  we observe that FSD-sgl achieves a significant improvement over QMIX, and FSD further outperforms FSD-wo-int, despite the latter already serving as a strong baseline. This improvement is particularly evident in terms of learning efficiency—if we limit the training steps to 1e6, the performance gain of FSD over FSD-wo-int becomes even more pronounced. **Overall, $Q_i^{int}$ does improve coordinated exploration.**
>
>   Besides, we also **add experiments to analyze how the coefficient $\alpha$ influences performance.** The results can be found in the ablation study section. In general, a coefficient between 0.01 and 0.1 can effectively improve exploration efficiency. In the relatively simple Predator-Prey scenario, while coordinated exploration is also crucial, $\alpha$ can take a larger value, such as 1, to further enhance exploration. However, on the more complex MMM2 map, setting $\alpha=1$ leads to excessive exploration by the agents, which ultimately reduces their learning efficiency.
>
>   **We also use a single-state matrix game to show that $Q_i^{int}$​​ is actually larger for state-action pairs with more fluctuating values.** The payoff matrix of the one-step game is as follows:
>
>   |      | A    | B    | C    |
>   | :--: | :--: | :--: | :--: |
>   | A    | 10   | -2   | -2   |
>   | B    | -2   | 0    | 0    |
>   | C    | -2   | 0    | 0    |
>
>   This symmetric matrix game has the optimal joint action (A, A), and captures a very simple cooperative multi-agent task, where we have two agents with three actions each. The learning results for FSD-VDN at time step 200 and time step 250 are as follows:
>
>
>   | time step 200 | 0.2169(A) | 0.0807(B) | -0.1377(C) |
>   | :-----------: | :-------: | :-------: | :--------: |
>   | 0.1612(A)     | 0.3780    | 0.2419    | 0.0235     |
>   | 0.0299(B)     | 0.2467    | 0.1106    | -0.1078    |
>   | -0.1719(C)    | 0.0450    | -0.0912   | -0.3095    |
>
>   | time step 250 | 1.9607(A) | -0.7665(B) | -0.7595(C) |
>   | :-----------: | :-------: | :--------: | :--------: |
>   | 1.8073(A)     | 3.7680    | 1.0408     | 1.0478     |
>   | -0.7835(B)    | 1.1773    | -1.5500    | -1.5429    |
>   | -0.7940(C)    | 1.1668    | -1.5605    | -1.5534    |
>
>   In the task, $\hat{Q}_i(s_0,a_A)$ for both agents is more fluctuating. The $Q_i^{int}$​​  at time step 250 is as follows:
>
>   | $Q_i^{int}$ at time step 250 |A|B|C|
>   | :--------------------------: | :------: | :----: | :----: |
>   | Agent_1                      | 18.8633 | 8.3257 | 7.0778 |
>   | Agent_2                      | 17.4624 | 8.3065 | 7.4410 |
>
>   $Q_i^{int}$ is indeed larger for $(s_0,a_A)$​ with more fluctuating values.
>
>   **We clarify that it is EMC that first points out that the prediction error against the individual Q-values can capture both the novelty of states and the interdependency among agents. Our contribution lies in providing an intuitive and detailed analysis of the prediction error.**

---

> ### Author Response · Authors · 2025-03-24
> **Response to reviewer z7h1 (3/3)**
>
> The critical issues are closely related to the weaknesses and we hope our response to the weaknesses has addressed your concern.  Here we briefly respond to the critical issues.
>
> - why and how the proposed method works
>
>   As pointed out by EMC, the prediction error against the individual Q-values can capture the novelty of states and interdependency among agents. FSD uses $Q_i^{int}$, a metric of the prediction error, as the exploration signal. As further analyzed in section 4.2, the larger prediction error results from fewer visits to the state-action pairs and stronger interdependency among agents. FDS biases agents to select actions with higher $Q_i^{int}$ to achieve coordinate exploration. Through coordinated exploration, agents can sample more valuable experiences that can boost learning efficiency and help agents to find a better policy. More experiments have been conducted, please refer to the response to the fifth weakness.
>
> - a detailed comparison with EXPODE
>
>   **The comparison has been conducted in the introduction where we compare FSD with previous curiosity-driven methods including EXPODE.** EXPODE bears the problem of non-stationary environmental dynamic and biased exploration direction. In contrast to EXPODE, FSD forgoes the explorer and considers the prediction error as a long-term exploration bonus $Q_i^{int}$, achieving farsighted self-directed exploration. Additionally, EXPODE and FSD both use clipped double Q-learning, but the intuition behind them is different. EXPODE uses clipped double Q-learning to construct a **$max$** target for generating intrinsic rewards and a **$min$** target as the TD target. FSD uses clipped double Q-learning to reduce noise in $Q_i^{int}$​ to make the signal more accurate.

---

### Review · Reviewer_PcAA · 2025-03-12

**Summary Of Contributions:**

This work presents a method for co-operative multi-agent exploration that uses prediction error in Q-values as a signal to guide exploration. Furthermore, the work incorporates a clipped double Q-Learning objective to reduce instability in the prediction error. Results show that the presented framework, referred to as Far-Sighted Direction (FSD), improves performance on popular MARL benchmarks such as Starcraft (SMAC) and Predator-Prey. Ablations show that both components (prediction bonus and double Q-Learning) contribute to the success of the method.

**Audience:**

Yes

**Claims And Evidence:**

Yes

**Requested Changes:**

1. The introduction dives too quickly into finer technical details and is difficult to follow. The overview of the proposed method should be simplified and some of the content should be moved to the Method section.  I think the controller and predictor terminologies need to be properly defined in the methods section. “Controller” and “Predictor” are not standard terminology and the section begins of talking about individual controller Q-values, without introducing proper notation for it.  [Strengthen the work]

2.  Is it possible to show results with larger number of agents? It seems that this approach might not scale well if multiple agents are learning in parallel due to increased non-stationarity. [Strengthen]

3. Why is method restricted to a Q-Mix type setup? If N individual agents are learning in an environment with a shared reward structure and individual Q-networks, shouldn’t it be possible to train using the method described in this paper? [Critical to answer]

4.  From the ablation experiments, it  seems that FSD-wo-int also performs extremely well and is only marginally below FSD. However, FSD-wo-int is simply using clipped double Q-Learning. This suggests that main gains of the experiments come from clipped double Q-Learning (which is arguably well-known) while the intrinsic reward only has a small contribution (even though it is the main focus of this paper). The authors should feel free to refute this as it is based on my understanding of the paper. [Critical]

5. Is the predictor network initialized to be the same as the controller? If not, is there a warm-up period without intrinsic rewards that could be useful to stabilize the Q-values because random initialization might lead to assignment of extremely noisy intrinsic rewards. [Strengthen]

6. The work mentions that considerable time is being saved by foregoing an explorer. Can the authors roughly estimate the increase in computational efficiency (through a metric such as wall clock time)? Even though the explorer is removed, the addition of a predictor along with maintaining 2 versions of the network might increase training time. [Critical to support claims]

7. It would be useful to have brief descriptions of the baseline algorithms, the current plots are not easy to decipher.

8.  How often is the predictor network updated? Updating too fast might reduce the signal while updating too slowly might lead to staleness in exploration. Is this controlled by some hyperparameter? [Strengthen understanding]

**Strengths And Weaknesses:**

Strengths:

1. The method presented is intuitive, simple to understand and builds upon existing curiosity-based methods that use prediction error to guide exploration.
2. Results presented on 2 popular MARL domains show that the method not only outperforms existing baselines, but is also more computationally efficient than some of the existing exploration methods.

Weaknesses:

1. The writing has scope for further improvement. The introduction is overly technical and dives into details that are better described in the Method section.
2. The novelty of the work is limited. One of the main contributions of the work is applying clipped double Q-Learning, which is an existing method that is already known to stabilize training. The primary contribution is the bonus based on prediction error, however the contribution of this component in improving performance is not very significant.
3. Some open questions remain, and answering them could improve the strength of the paper.

---

> ### Author Response · Authors · 2025-03-24
> **Response to reviewer PcAA (1/2)**
>
> We appreciate your time in reviewing our paper and your helpful suggestions. Please find our response below and we have revised the manuscript accordingly, marking the changes in red
>
> - The introduction dives too quickly into finer technical details and is difficult to follow.
>
>   Thank you for your suggestions on the writing of the Introduction section. We provide a relatively detailed description of our method in the Introduction section mainly to highlight its differences from previous approaches. In the Method section, we elaborate on both the controller and the predictor, and **in the "To address the limitation" paragraph of the Introduction, we not only discuss the shortcomings of previous work but also clarify the meaning of controller and predictor.**
>
>   **We have rephrased the paragraph to make it more explicit**, and we hope this addresses your concern. **The revised version is as follows:**
>
>   To address the limitation, the intrinsic reward is designed as the prediction error of the individual Q-values for value decomposition methods, promoting efficient coordinated exploration. To avoid excessive coupling between intrinsic reward generation and policy learning, these curiosity-driven methods typically employ a controller-explorer-predictor structure where the controller learns the policy used to interact with the environment and the explorer is only used during training to generate the intrinsic reward. The prediction error between the explorer and the preditor is used as the intrinsic reward for the controller to enhance exploration...
>
> - results with larger number of agents
>
>   We evaluated the performance of our method on Predator-Prey with a larger number of agents and VDN is selected as the backbone. First, we double the number of agents to **16** and FSD-VDN can still quickly find the optimal policy to succeed. Further we increase the number of agents to **32**, FSD-VDN still succeeds in the task. **We have added the results to the appendix.**
>
>   **We adopted the CTDE framework and the centralized training alleviates the non-stationarity problem**. However, the increasing number of agents indeed makes coordination more challenging and makes it more difficult to learn the optimal policy.
>
> - Why is the method restricted to a Q-Mix type setup?
>
>   Thank you for noticing and pointing this out. The method proposed in this paper can indeed be used to enhance the performance of approaches that follow the shared reward structure and individual Q-networks setup.  In fact, as mentioned in the Method section, **“It’s worth noting that our method can be embedded into any value decomposition method"**(including VDN, QMIX, QPLEX, and so on).
>
> - From the ablation experiments, it suggests that main gains of the experiments come from clipped double Q-Learning.
>
>   On the MMM2 and 6h_vs_8z maps, FSD-wo-int performs quite well. However, we also observe that FSD-sgl achieves a significant improvement over QMIX, and FSD further outperforms FSD-wo-int, despite the latter already serving as a strong baseline. **This improvement is particularly evident in terms of learning efficiency—if we limit the training steps to 1e6, the performance gain of FSD over FSD-wo-int becomes even more pronounced.**
>
>   **Additionally, we add new experiment results on Predator Prey for the ablation study.** On Predator Prey, FSD-wo-in does not show any improvement over QMIX and similarly fails on the Predator-Prey (PP) task. In contrast, FSD-sgl succeeds in the PP task and achieves indistinguishable performance from FSD. **This suggests that, in relatively simpler scenarios where the overestimation issue is less severe and also less noise in $Q_i^{int}$, clipped double Q-learning does not have a significant impact. On the other hand, $Q_i^{int}$ can help find the optimal policy by directly enhancing coordinated exploration.**
>
>   Across both the complex SMAC tasks and the simpler Predator Prey task, combining clipped double Q-learning with $Q_i^{int}$ maximizes the effect of $Q_i^{int}$, more accurately directing agents in coordinated exploration and achieving high performance stably.
>
> - Is the predictor network initialized to be the same as the controller?
>
>   **The predictor network is not initialized to be the same as the controller.** The intrinsic signal can be indeed noisy at the beginning of the training. However,  **we adopt $\epsilon-$ greedy after $Q_i^{act}$ and  $\epsilon$ is also large at the beginning**($\epsilon$ starts at 1 and decreases to 0.05 as training processes). Thus, the exploration at the very beginning is quite random and the noisy exploration signal will have little impact on it.

---

> ### Author Response · Authors · 2025-03-24
> **Response to reviewer PcAA (2/2)**
>
> - Can the authors roughly estimate the increase in computational efficiency (through a metric such as wall clock time)?
>
>   Here we roughly estimate the increase in computational efficiency. Before we start, **we clarify that the predictor also exists in previous works.** We have rephrased the related description in the introduction section to make it more explicit. **We have conducted experiments on the MMM2 map of SMAC and recorded the time required for FSD, EXPODE, and EMC to update the network once (calculated as the average time over 1,000 network updates). The results show that FSD takes 0.027s, EXPODE takes 0.073s, and EMC takes 0.43s.** The time cost for FSD is significantly shorter than that of both EXPODE and EMC. The extremely high time cost for EMC is due to its use of episodic Q-learning. After episodic memory is removed, the update time for EMC(without episodic memory) is reduced to 0.056s, yet it is still longer than that of FSD.
>
> - It would be useful to have brief descriptions of the baseline algorithms
>
>   **We have added an introduction to Weighted-QMIX and QPLEX in the Background section.** The description is as follows:
>
>   Subsequently, Weighted-QMIX introduces weighting into Q-value learning and places more importance on better joint actions to address the limitation of monotonicity. More algorithms have been proposed to enhance representation expressiveness, such as QTRAN, QPLEX and DAVE.
>
> - How often is the predictor network updated?
>
>   **The Predictor network is trained synchronously with the Controller network**, meaning that every time we update the controller network, we also update the Predictor network. The prediction error between the predictor and controller is used to determine **currently** which state-action pairs require further exploration. Therefore, their updates are synchronized.

---

### Decision · Action_Editor_HP73 · 2025-04-11

**Recommendation:** Accept with minor revision

**Comment:**

After considering the reviews and the authors’ rebuttal, I recommend acceptance of this paper. All reviewers now lean toward acceptance, acknowledging that the proposed method—Farsighted Self-Direction (FSD)—is intuitive, performs well empirically, and removes the need for an explicit explorer in MARL.

While initial concerns were raised about the clarity of the method and the attribution of performance gains to clipped double Q-learning, the authors’ responses and additional experiments helped address these points to a satisfactory degree.

For the final revision, the authors should aim to clarify terminology and improve notation throughout the paper. Some figures, particularly Figure 5, would benefit from better readability, and a more thorough comparison with prior work, such as EXPODE, would further strengthen the contribution.

**Audience:**

Yes, some individuals in TMLR’s audience would be interested in the paper. It addresses coordinated exploration in MARL, a relevant problem, and proposes a novel method that could appeal to those working on intrinsic motivation or efficient exploration techniques.

**Claims And Evidence:**

Yes, the claims in the submission are generally supported by empirical results, showing that the proposed method improves performance on standard benchmarks.